# A qualitative study on the reasons for solitary eating habits of older adults living with family

Kyo Takahashi[1,2]*, Hiroshi Murayama[2], Tomoki Tanaka[2], Mai Takase[2],
Unyaporn Suthutvoravut[2], Katsuya Iijima[2]

**1** Department of Public Health, School of Medicine, Dokkyo Medical University, Mibu, Japan, **2** Institute of Gerontology, The University of Tokyo, Tokyo, Japan

* tkyo-tky@umin.ac.jp

**Data Availability Statement:** The data in this study is restricted because the ethical approval was not sought for public data sharing from the Ethics Committee at the University of Tokyo and the

## Abstract

Eating alone while living with family members is a risk factor for mental health decline in old age. However, little is known as to why older adults choose to eat alone, even with family present. This study therefore aimed to explore reasons for older adults eating alone despite living with family members, using a qualitative approach. Fifteen people aged 65 years and older (11 men and 4 women) who were eating alone while living with family members were included in the study. These individuals were selected from the participants of the Kashiwa cohort study conducted in 2016. Individual interviews were conducted using an open-ended format. All interviews were recorded and transcribed. The data were further thematically analyzed using a qualitative software package, NVivo 11. We extracted six themes as reasons for eating alone and hypothesized interactions among these themes. The extracted themes were: "age-related changes," "solo-friendly environment," "family structure changes," "time lag for eating," "bad relationships with family members" and "routinization." To assess interactions, the themes were categorized as "background factors," "triggers," and "stabilizers." The aforementioned themes could lead to the development and sustained behavior of eating alone among older adults living with family members. As most themes describe conditions that are likely to remain static, it may not be realistic to encourage such individuals to begin eating with family members. The promotion of meals with neighbors or friends could be effective in alleviating the negative consequences of eating alone.

## Introduction

As people age, the chance of them eating alone increases. In Japan, where the aging rate exceeds 28.0%, the proportion of older adults living alone has increased from 4.3% to 15.5% in men and from 11.2% to 22.4% in women in the last four decades [1]. This demographic trend could result in an increase of the proportion of older adults eating alone [2].

Whether they eat alone or with others can impact older adults' health and well-being. Several studies have revealed the negative effects of eating alone. For example, eating alone is associated with a decreased quality and quantity of food intake [3, 4], oral frailty [5], and

participants were not informed of possible public data sharing when they provided informed consent. However, data can be made available from a non-author of contact at Institute of Gerontology, the University of Tokyo (contact via info.frail@iog.u-tokyo.ac.jp) for researchers who meet the criteria for access to confidential data.

**Funding:** This work was supported by a Japan Society for the Promotion of Science (JSPS) KAKENHI Grant-in-Aid for Scientific Research (https://www.jsps.go.jp/english/e-grants/index. html) [Grant numbers: 15K08728, 19K21579] to KI and KT. The funders had no role in study design, data collection and analysis, decision to publish, or preparation of the manuscript.

**Competing interests:** The authors have declared that no competing interests exist.

depressive symptoms [6] among older adults. Eating alone appears to be a simple state; however, it can lead to people eating poorly, resulting in many unhealthy conditions.

Older adults' living status could also amplify the negative impact of eating alone. For instance, for older adults who eat alone despite living with family members, the health consequences appear to be dire. A series of Japanese cross-sectional studies observed an association between eating alone (despite living with family members) and frailty [7] as well as depressive symptoms among community-dwelling older adults [8]. Another longitudinal study revealed a higher mortality risk for older adults who eat alone than for those who do not, even if they live with family members [9].

While the negative impact of eating alone when others are present is well-established, the reasons for older adults choosing to do so are unclear. Understanding and documenting the actual reasons for this behavior could help provide insights into strategies for alleviating the negative impacts of eating alone in old age. Further, it could contribute to addressing underlying behavior issues related to eating alone. Therefore, this study explored the reasons why some older adults choose to eat alone despite living with family members.

## Methods

### Ethical approval

After detailed information related to the study was provided, all participants provided written informed consent. Approval of the study design was provided by the University of Tokyo Ethics Committee (15–103, 17–20).

### Participants

The sampling procedure is shown in Fig 1. A total of 51 adults aged 65 and older who had reported that they regularly ate alone while living with family members were initially recruited from a previous cohort study conducted from September to October, 2016 in Kashiwa city, Japan. Participants were mailed solicitations, and 34 replies were received. Of these, 14 declined to participate owing to unstable health conditions, inaccessibility to the interview location, or a change in their eating or living situation. Thus, a sample of 20 participants were interviewed; however, five were excluded from the analysis because they did not currently eat alone, resulting in a final study sample of 15 people.

### Data collection

From August 2017 to March 2018, we conducted semi-structured interviews with all participants in a private room of the Institute of Gerontology at the University of Tokyo. All participants arranged their own transportation to the site. The overall concept of the study was explained, and informed consent addressed prior to the interviews. One or two researchers conducted interviews with each individual participant using an interview guide that included open-ended questions regarding recent eating habits, family history, and social status (See S1 File). According to the flow of the conversation, questions were asked in a random order. The researchers took field notes to record the participants' non-verbal behaviors. The interview times varied between 33 and 70 minutes. All conversations were recorded and transcribed afterwards.

### Data analysis

We inductively analyzed the verbatim transcripts of the 15 participants [10]. First, we familiarized ourselves with the qualitative data by reviewing the field notes, listening to the recorded

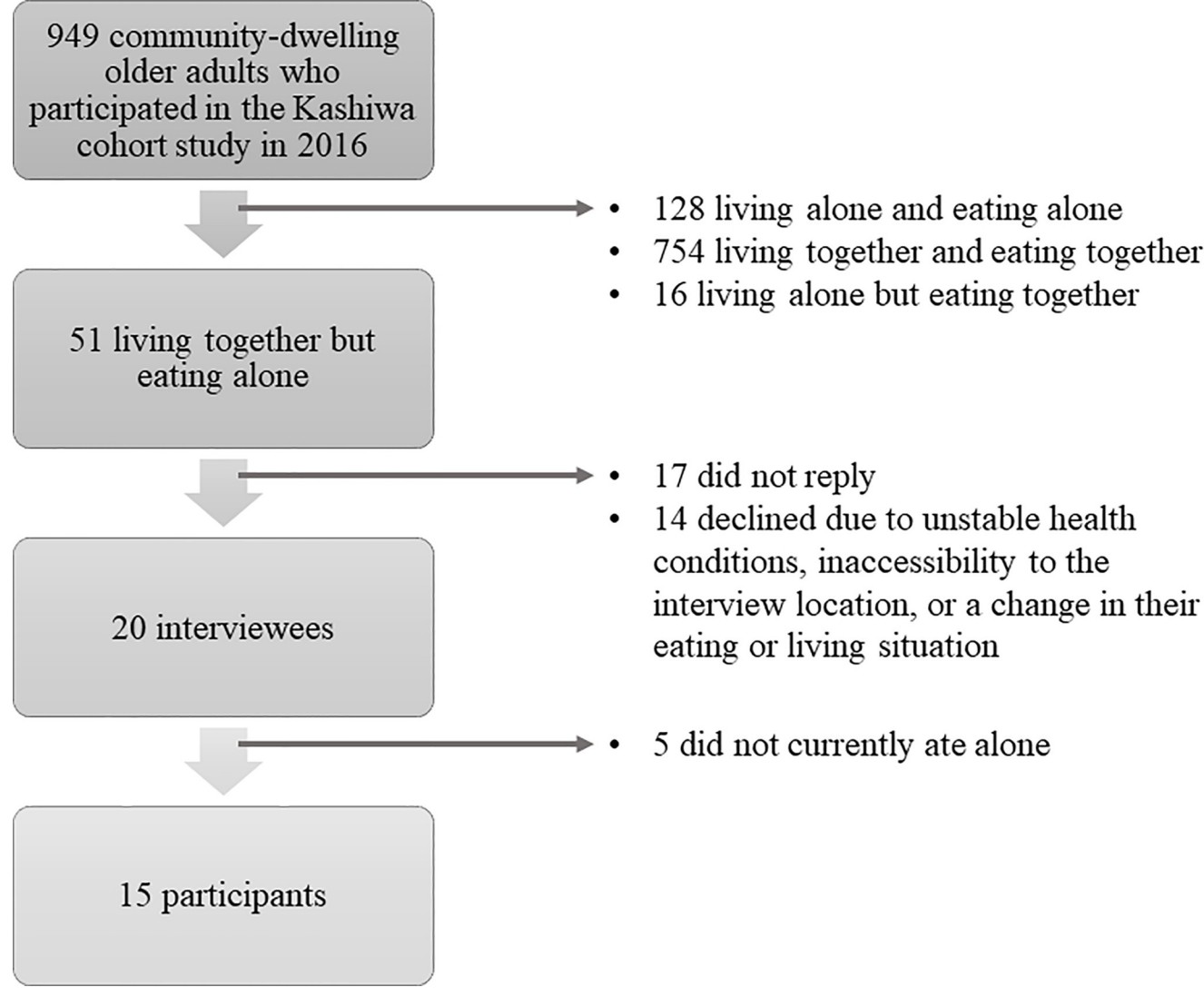

**Fig 1. Sampling procedure.**

interviews, and repeatedly reading the transcripts. Next, the first author initially extracted tentative themes as groups of salient phrases. We then discussed and developed definite themes and produced a conceptual model to illustrate any thematic relationships. During this process, the labels of themes were changed repeatedly, until we reached consensus on the labels that correctly identified the themes. During the data collection, we continuously analyzed the data. Moreover, in the closing stages of data collection/analysis, we ensured that no new themes were extracted [11]. To enhance the validity of the analysis, we presented and discussed the results with interdisciplinary experts in gerontology at the University of Tokyo. A qualitative software package, NVivo 11 (QSR International Pty Ltd, Victoria), was used owing to its versatility for thematic analysis.

## Results

Characteristics for all 15 participants are shown in Table 1. Eleven participants were men and the mean age was 77.3 years old (range: 70–85). Eleven of the participants lived only with a

Table 1. Participant characteristics.

| Participant ID | Family member(s) living with participant | Number of years of eating alone | Psychological status[a] | Trigger(s) of eating alone |
|---|---|---|---|---|
| A | Son | 15 | 2 | Family structure changes Time lag for eating |
| B | Spouse | 7–8 | 3 | Bad relationships with family members |
| C | Younger brother and sister-in-law | 17 | 4 | Family structure changes |
| D | Spouse and daughter | 20 | NA | Time lag for eating |
| E | Son | 25 | 5 | Family structure changes Time lag for eating |
| F | Spouse and son | 5–6 | 2 | Time lag for eating |
| G | Spouse and son | 10 | 5 | Family structure changes Time lag for eating |
| H | Spouse | 4 | 6 | Family structure changes Time lag for eating Bad relationships with family members |
| I | Spouse | 30 | 12 | Family structure changes Bad relationships with family members |
| J | Son | 13 | 1 | Family structure changes Time lag for eating |
| K | Son | 22 | 3 | Time lag for eating Bad relationships with family members |
| L | Spouse | 20 | 13 | Time lag for eating |
| M | Spouse | 10 | 12 | Time lag for eating Bad relationships with family members |
| N | Son | 18 | 5 | Time lag for eating |
| O | Spouse | 20 | 2 | Time lag for eating |

[a]Scores of geriatric depression scale are shown. Higher scores indicate more depressive symptoms.

spouse or child(ren) and the average number of years the participants had been eating alone was 15.8 (range: 4–30). The participants' psychological statuses were measured using the geriatric depression scale 15, and only three participants had scores of 10 or more, indicating "severe depression."[12] We extracted six themes that represent the broad categories of reasons for older people eating alone at home: "age-related changes," "solo-friendly environment," "family structure changes," "time lag for eating," "bad relationships with family members," and "routinization." Each theme's characteristics and representative narratives are shown below.

## Reasons for eating alone despite living with family members

**Age-related changes.** Every participant reported experiencing various age-related changes, such as the loss of a social role, a decline in physical strength, or changes in food preferences and tastes. For instance, Participant A took their medicine at a specific time, which resulted in time lag for eating with their son.

*My son comes home around 8:30 (pm) and he eats dinner by himself then. But me, because of this taking medicine, I eat earlier. I eat around 5:30 (pm). Then, I take the medicines around 6:00 (pm).*

Family members of participants also experienced age-related changes. For example, Participant O's spouse used to cook, and they ate together. However, their roles changed completely after retirement, resulting in them eating alone.

*My spouse doesn't cook. After I retired, there is no hope. My spouse doesn't take care of me, and I also don't feel uncomfortable with that. Little by little, I have come to prefer cooking by*

*myself. Before retirement, I was busy and didn't have time (to cook). But now, I have time enough and to spare. My spouse did cook before, but now, I do it almost every time.*

**Solo-friendly environment.**   Some participants periodically used food delivery services, purchased a single serving of food at a nearby store, or cooked a frozen dinner designed for one. These convenience foods are conducive to eating alone. As an example, Participant K regularly used convenience food.

*When I return home, I don't want to cook. Then, I eat food that is in the house, like dumplings, frozen dumplings, or something like that. They can be easily cooked in 10 or 15 minutes. I eat that kind of food. And sometimes, I eat canned food. My friend also says she doesn't like to cook for her husband. I understand that because we are almost 80. Everyone says so. But I'm easy because I eat alone (although I have a son living with me). I eat some noodles or spaghetti for my dinner, or I buy a pizza nearby.*

**Family structure changes.**   The results showed that often, the family structure changed due to the deaths of certain family members or children becoming more independent. Such changes forced seven participants to eat alone, even if they did not wish to do so. For instance, Participant A lost their spouse 15 years prior to the study and they have eaten alone ever since, despite living with their son. They stated:

*It's been 15 years since my spouse died; I was about 70. In the beginning, I sometimes had dinner with my son. But gradually, maybe because we are both men, we (came to) eat separately. He and I are somehow shy. He talked with me while my spouse was alive, but now he doesn't even say "I'm home." Yes, I've eaten alone since that time.*

**Time lag for eating.**   Another reason for eating alone was the mismatch in mealtimes between family members. If either a participant or their family member(s) was socially active, their eating times were less likely to coincide. Among the 11 participants who had this theme, eight were living with adult children who worked full-time. For instance, Participant K reported that eating times differed for their son and themselves, and they also had different food preferences.

*"My son was back from abroad, and we have lived together since then. But we eat separately. This is because our schedules are different, and what he likes is different from what I like. We know we are different. So, we eat independently."*

## Bad relationships with family members

Five participants had strained relationships with certain family members for various reasons. This theme was mainly observed among participants living with their spouses. Participants reporting this theme seemed to be satisfied with eating alone. For them, eating alone was viewed as a relief from psychological stress. For example, Participant I enjoyed eating without their spouse, as their spouse has issues with alcohol. Participant I stated:

*Actually, I don't want to say this. . . My spouse has (had an) alcohol dependency for a long time. My spouse is so nagging and always complains a lot. I cannot stand it. Thus, I prefer*

*eating alone. When my daughter was little, we were eating together. But I don't remember from when (we began to eat separately). Why? Because I felt so bad and we fought each time we ate together.*

**Routinization.**   When the participants repeatedly ate alone, this behavior became a standard routine in their daily living. Once they were habituated to eating alone, participants did not seem to mind it. For instance, Participant H and their spouse had eaten separately for a long time as a form of respect.

*We don't care (about mealtimes) at all. It would bother me if we did the impossible and ate together. I'm not avoiding eating together. We just don't care.*

## Discussion

We extracted six themes from the reasons for older adults eating alone despite living with family members. As shown in Fig 2, we categorized two themes as "background factors" that were predictors of eating alone, three as "triggers" that directly lead to eating alone, and the remaining one as a "stabilizer" that reinforces a tendency to eat alone. All participants reported a trigger that led them to eating alone and most mentioned a background factor or a stabilizer that helped them to habituate to the behavior of eating alone.

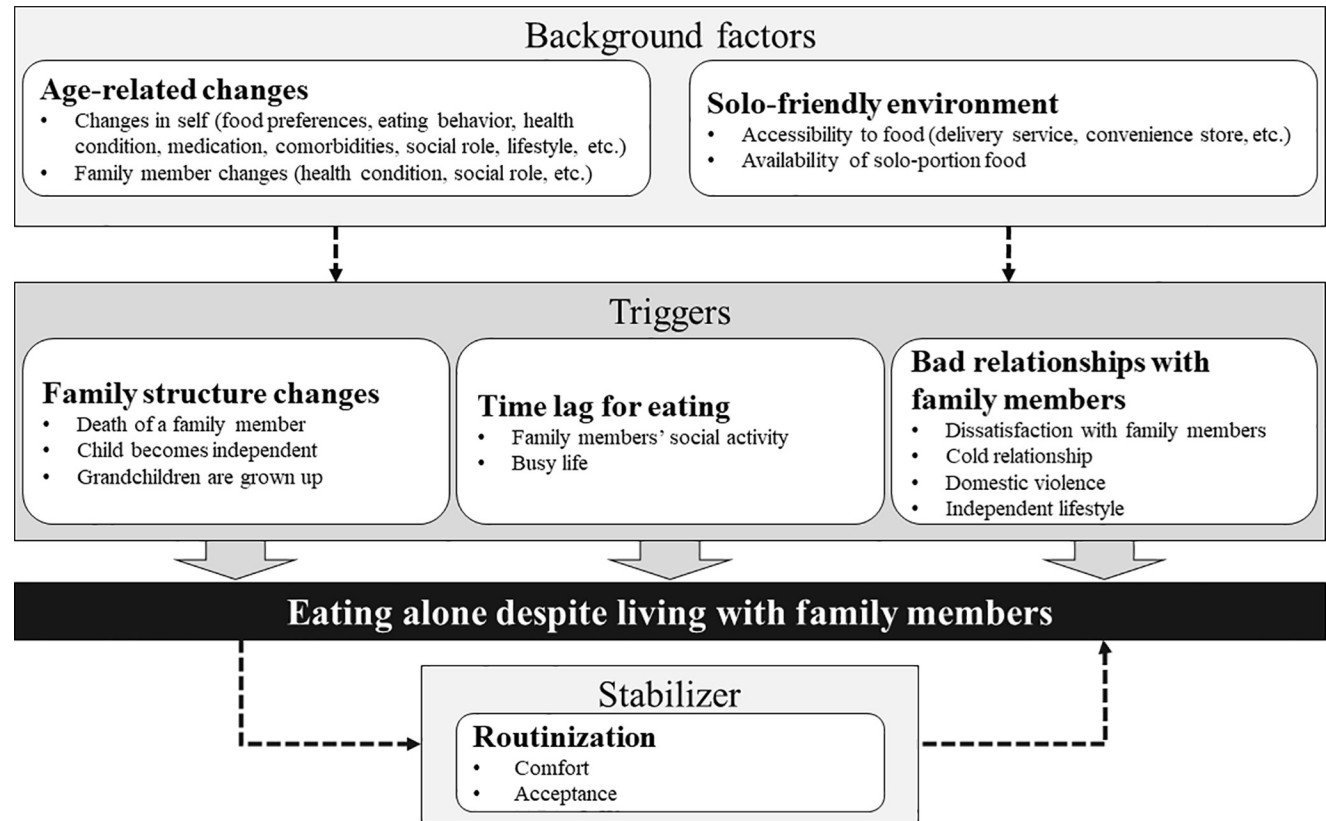

**Fig 2. Background factors, triggers, and stabilizer for eating alone despite living with family members.**

## Background factors

The themes "Age-related changes" and "solo-friendly environment" were categorized as background factors which could be predictors of eating alone despite living with family members. First, age-related changes occur in both individuals and their family members. For example, the quality and quantity of food we desire tends to change as we age [3, 13]. This difference in food preferences between older adults and family members might result in separate meal preparation and intake behaviors. Another example is family members' health conditions. If either the older adult or a family member becomes disabled due to aging, it would affect their relationships and even lifestyles, such as their eating behavior.

Furthermore, environments become more solo-friendly as the number of those living alone increase. For individuals living alone (one-third of the Japanese population), eating alone is habitual and commonplace [14]. In such environments, food delivery services are desirable, preserved foods are becoming tastier and more nutritious, and convenience stores are increasingly more accessible [15]. These conditions enable those eating alone to become comfortable with the practice.

## Triggers

Changes in family structure, mealtime mismatches, and bad relationships with family members can be considered as triggers. Of these, family structure changes seemed to be the main trigger for eating alone. Family structures often change as people age [16]. Several decades ago, many generations within a single family lived together in Japanese societies. However, nuclearization of the family has rapidly advanced in recent years. For instance, in 2016, more than 50% of older adults lived alone or only with a spouse [1]. This decrease in the number of cohabiting family members could directly or indirectly lead to a greater number of people eating alone. This situation may be beyond an individual's control and can lead to meals taken alone, even if company is desired.

The eating-time lag trigger was mostly observed between parents and their adult children. Participants' children tend to be more socially active than they are and—as most of the participants live in the suburbs of Tokyo—their adult children living at home typically have long commutes to work. The resulting pattern of children leaving early in the day and arriving home late in the evening creates circumstances where mealtimes for different members of a household naturally vary. Regardless of whether participants desire to eat with their families or alone, time lag for eating led to them eating alone.

Poor relationships with family members as a trigger was often reflected between spouses. Due to domestic violence, fatigue of the marriage relationship, or lifestyle differences between spouses, eating with one's spouse was often reported as a stressful daily event. To avoid this stress, participants preferred to eat alone. Family relationships can be destroyed by lasting and accumulated dissatisfaction towards a family member, resulting in various behaviors, including eating alone.

## Stabilizer

Routinization was categorized as a stabilizer of eating alone. For many participants, eating alone has just become a standard daily activity. Therefore, participants did not feel that their eating behavior was particularly special or remarkable. This routinization could result in strengthening the behavior and being a barrier to intervention.

While most of the themes identified by participants would likely be difficult to change, interventions could be implemented to disrupt repetitive habits and reduce the occurrence of solo-friendly eating environments. For example, building community-friendly eating

environments in public spaces could limit older adults' desire to eat alone. In Japan, a community activity for vulnerable children, *kodomo-shokudo* (dining for children), was a very popular social movement [17]. This activity provided a warm meal and place for latchkey children to interact with others and avoid isolation. Recently, a similar activity for older adults was established in some areas across Japan [18]. Eating together could improve older adults' nutrition statuses and even their subjective well-being, regardless of who the companions are [19]. A hospital-based study reported the positive effect of eating together on increasing energy intake among 48 older patients [20]. Another cross-sectional study revealed the negative effect of a low frequency of eating together on health outcomes among older adults living alone [21]. Although such studies are accumulating rapidly, more focus should be placed upon increasing the opportunities for older adults to eat together to reduce the risks of depression, malnutrition, and mental decline.

A few limitations of this study should be noted. First, the small sample size could limit the generalizability of our findings. Additionally, our small sample of older adults eating alone—but not living alone—may not be representative of most Japanese older adults in these types of situations. For instance, previous research shows that older adults eating alone while living with family members tend to report significant symptoms of depression [8, 9]. However, many of our participants seemed to be psychologically healthy, as most of their scores on the geriatric depression scale were low. Moreover, those eating alone and distressed by this fact are likely to be less willing to participate in such a study. Future studies should find ways to diversify the sample pool. Second, data collection was not conducted using snowball sampling. Our participants were selected from the participant group of an existing population-based study. This nested study design limited our ability to recruit more participants. Considering the themes, although no new themes were extracted in the closing stages of data collection/analysis, recruiting more participants could enrich the study results. Another limitation is that no information was collected from participants' family members. Family participation could have provided alternative explanations for why our participants ate alone, providing greater insight into the conditions and circumstances. Lastly, the participants' histories of changes in eating behaviors could not be accounted for in the present study. For instance, recent articles revealed a universal trend of people spending less time eating meals [22, 23]. The present generation would think lightly of eating at home because they are busier with daily tasks than the past generation. Thus, there could be cultural and historical factors influencing the tendency of older adults to eat alone, which may not be easily quantified.

## Conclusions

The present study illustrated reasons why older adults are eating alone despite living with family members. Eating alone could manifest due to changes in the family structure, a mismatch in mealtimes between members of the family, and poor family relationships. Eating alone can then be further habituated by the advent of eating environments that are accommodating to solo eating. Furthermore, basic age-related shifts in the types of food and quantities that individuals prefer could affect previously mentioned reasons for eating alone despite living with family members. These findings could be useful for promoting interventions for getting older adults to eat in community settings. Eating alone is not an inevitable and intractable proposition for many older adults. Thus, while eating with family members could be difficult to achieve, encouraging older adults to perhaps take meals with neighbors and friends could be promising. The results of this study are sufficient to warrant larger studies using similar methods to continue the examination of the circumstances and conditions of older adults eating alone.

## Supporting information

**S1 File.**
(DOC)

## Acknowledgments

The authors are deeply thankful for the participants who participated in this study.

## Author Contributions

**Conceptualization:** Kyo Takahashi, Katsuya Iijima.

**Data curation:** Kyo Takahashi.

**Formal analysis:** Kyo Takahashi, Hiroshi Murayama, Tomoki Tanaka, Mai Takase, Unyaporn Suthutvoravut.

**Funding acquisition:** Kyo Takahashi, Katsuya Iijima.

**Investigation:** Kyo Takahashi, Hiroshi Murayama, Tomoki Tanaka, Mai Takase.

**Project administration:** Kyo Takahashi, Tomoki Tanaka.

**Supervision:** Katsuya Iijima.

**Validation:** Mai Takase.

**Visualization:** Kyo Takahashi, Hiroshi Murayama, Mai Takase.

**Writing – original draft:** Kyo Takahashi, Hiroshi Murayama, Tomoki Tanaka, Mai Takase, Unyaporn Suthutvoravut, Katsuya Iijima.

**Writing – review & editing:** Kyo Takahashi, Hiroshi Murayama, Tomoki Tanaka, Mai Takase, Unyaporn Suthutvoravut, Katsuya Iijima.

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
