## [Decision Letter · Decision Letter 0]

22 Jan 2020

PONE-D-19-29527

A qualitative study on the behavioral reasons for solitary eating habits of community-dwelling older adults living with family

PLOS ONE

Dear Dr. Takahashi,

Thank you for submitting your manuscript to PLOS ONE. After careful consideration, we feel that it has merit but does not fully meet PLOS ONE’s publication criteria as it currently stands. Therefore, we invite you to submit a revised version of the manuscript that addresses the points raised during the review process.

The reviewers have raised concerns in a couple of areas -- more detail is needed on the methods, particularly how you made determinations in the qualitative analyses. There are some additional citations to help you as you revise your manuscript.

We would appreciate receiving your revised manuscript by Mar 07 2020 11:59PM. To enhance the reproducibility of your results, we recommend that if applicable you deposit your laboratory protocols in protocols.io, where a protocol can be assigned its own identifier (DOI) such that it can be cited independently in the future. For instructions see: http://journals.plos.org/plosone/s/submission-guidelines#loc-laboratory-protocols

We look forward to receiving your revised manuscript.

Kind regards,

Heidi H Ewen

Academic Editor

PLOS ONE

2. In Table 1 please remove sex, age and replace "wife"/"husband" with spouse as these information could be potentially identifying.

Reviewers' comments:

Reviewer's Responses to Questions

**Comments to the Author**

1. Is the manuscript technically sound, and do the data support the conclusions?

Reviewer #1: Yes

Reviewer #2: Partly

2. Has the statistical analysis been performed appropriately and rigorously? 

Reviewer #1: N/A

Reviewer #2: I Don't Know

3. Have the authors made all data underlying the findings in their manuscript fully available?

Reviewer #1: Yes

Reviewer #2: No

4. Is the manuscript presented in an intelligible fashion and written in standard English?

Reviewer #1: Yes

Reviewer #2: Yes

5. Review Comments to the Author

Reviewer #1: The study addresses very important issue on eating alone, focusing on the elderly eat alone while living with family members. It is interesting that the article explores the reasons underlying eating alone in a qualitative study. However, there are a few points to be explained detail.

1. The article might need more discussion about the differences of family members living together. The situation might be different whether the elderly living with spouse or with children (especially with the category “Time-rag for eating”).

2. I could not understand why the authors set the category of “Routinization” apart from “Eating in solo-friendly environment”. Routinization seems more like a result of all the other reasons and itself is the situation of eating alone (eating alone routinely). This paragraph might needs more discussion and example.

Reviewer #2: Dear Authors,

Although I believe that the topic is very relevant, mayor revisions need to take place in order to improve, the introduction, the methods and discussion section. I would also suggest additional analyses.

- introduction: please focus on the topic, the first paragraph is not about eating alone and should be deleted. Start with paragraph 2 (line 48) and only focus on older adults (your target population). Please explain in depth what is known from literature about the reasons people eat alone, what kind of research and what are the consequences. What are the mechanisms behind the presumed relation between eating alone and decreased/insufficient food intake and mental health. Do we know if eating alone when living together is worse compared to living alone completely? do we know if the people that choose to eat alone are better off compared to people that are forced to eat alone?

- Methods: Please add more information and details on the sample population. Where did they come from, what were reasons to not participate? please add the semi-structured interview guideline. And more details on how you guided the interview. how many researchers were present during the interview? where is all info collected - add more extended table 1? Not all in table 1? Table 1: what is 'going out'? how were the questions asked? add answer categories if used..

Update the part on data analyses with more details on the data handling and data saturation, how do you know that the number of interviews were enough to ensure that no new themes would emerge from additional data.

- results: Please add more information on: who mentioned the specific themes, where answers given mainly by men or women, older younger older adults, where the quotes by older adults that choose to eat alone, or were forced to do so? Add analayses and cross-analyses, compare groups.

For the themes: eating in solo-friendly, age related please provide other quotes and examples that really reveal the link between eating alone and the health consequence. Please extent figure 1 with more details.

- discussion please elaborate to the part about stabilizers? it seems like this is a good thing, while you explain in the intro that eating alone has negative health consequences. Or is this different for people that are used to living alone/eating alone?

217-221: was this a result from the interviews or is this a solution if there is no other choice?

What is the solution for the effect on mental health? eating together in community setting? please support with references and studies that show that this might help/improve mental status / quality of life?

234-237 support with more studies - RCTs available?

242-243 how can you conclude physiologically healthy participants? If you have this information, more details should be included in demographics and results section.

248-251: do people eat more when they take more time for their meals? please explain about study mentioned and support with other references.

263-265 this contradict the part on generalizability (lines 238/239) we don't know enough about the sample group to state something on generalizability or extension of results to other older adults in Japan.

Please critically review the manuscript and improve thoroughly so that this interesting topic is described more in detail and researchers and health care professionals lean how important it is to eat together! Now the scientific background and scientific interpretation is too minimal.

Please use the following references for qualitative studies that include much more information on data handling, saturation and description of qualitative data:

Pope C, Ziebland S, Mays N. Qualitative research in health care. Analysing

qualitative data. BMJ (Clinical research ed). 2000;320(7227):114-6.

Guest G, Bunce A, Johnson L. How many interviews are enough?: An experiment with data saturation and variability. Field Methods. 2006;18(1):59-82.

McLafferty I. Focus group interviews as a data collecting strategy. Journal of

Advanced Nursing. 2004;48(2):187-94.

Perspectives on the Causes of Undernutrition of Community-Dwelling Older Adults: A Qualitative Study. van der Pols-Vijlbrief R, Wijnhoven HAH, Visser M. J Nutr Health Aging. 2017;21(10):1200-1209. doi: 10.1007/s12603-017-0872-9.

Perceptions on the use of pricing strategies to stimulate healthy eating among residents of deprived neighbourhoods: a focus group study. Waterlander WE, de Mul A, Schuit AJ, Seidell JC, Steenhuis IH. Int J Behav Nutr Phys Act. 2010 May 19;7:44. doi: 10.1186/1479-5868-7-44.

6. PLOS authors have the option to publish the peer review history of their article (what does this mean?). If published, this will include your full peer review and any attached files.

Reviewer #1: No

Reviewer #2: No

---

## [Author Response · Author response to Decision Letter 0]

15 Mar 2020

Dear Dr. Heidi H Ewen and reviewers,

We would like to express our heartfelt appreciation for the insightful comments, constructive suggestions, and helpful information provided by the academic editor and the reviewers. According to these comments, we have carefully studied and revised our original manuscript. Please find our responses below.

Thank you very much for your consideration for the publication of our manuscript in PLOS ONE. We would be very grateful if you could give us further comments on the revised version of our manuscript.

Journal requirements and response

Response: We have ensured that our manuscript meets the journal’s style requirements.

2. In Table 1 please remove sex, age and replace "wife"/"husband" with spouse as these information could be potentially identifying.

Response: Based on your suggestion, we have removed the information of sex and age and replaced the expression “wife/husband” with “spouse” throughout the manuscript.

Response: We added this information in the ethics statement in the Methods section and in the online submission information: “After detailed information related to the study was provided, all participants provided written informed consent.” (p. 4, lines 62–63)

4. In your Data Availability statement, you have not specified where the minimal data set underlying the results described in your manuscript can be found. PLOS defines a study's minimal data set as the underlying data used to reach the conclusions drawn in the manuscript and any additional data required to replicate the reported study findings in their entirety. All PLOS journals require that the minimal data set be made fully available.

Response: Thank you for this observation. The data in this study cannot be shared publicly because the participants were not informed of possible public data sharing when they provided informed consent. However, data can be made available from Institute of Gerontology, the University of Tokyo (contact via info.frail@iog.u-tokyo.ac.jp) for researchers who meet the criteria for access to confidential data. We added this information to the online submission information.

Reviewer 1’s comments and responses

1. The article might need more discussion about the differences of family members living together. The situation might be different whether the elderly living with spouse or with children (especially with the category “Time-rag for eating”).

Response: Thank you for your suggestion. We agree that the situation would differ depending on whether older adults are living with a spouse or with children. We updated table 1 with information of the extracted themes (trigger of eating alone). Using that data, we explored a common tendency among those mentioned in a particular theme. As a result, “time lag for eating” was observed as the main reason in those living with their child(ren), as you pointed out. Regarding “bad relationships with family members,” this tendency was mainly observed in those living with a spouse. We added this information in the Results section (page 7, table 1; p. 10, line 161-162; p. 11, line 171).

2. I could not understand why the authors set the category of “Routinization” apart from “Eating in solo-friendly environment”. Routinization seems more like a result of all the other reasons and itself is the situation of eating alone (eating alone routinely). This paragraph might need more discussion and example.

Response: Thank you very much for your insightful observation. We discussed the interaction of themes again and reached the conclusion that there is a necessity of changing the conceptual model. “Eating in solo-friendly environment” was recategorized from “stabilizer” to “background factor.” Further, as you indicated, “routinization” was a result of eating alone, which could stabilize such behavior. Please see figure 2.

Reviewer 2’s comments and responses

1. Introduction: please focus on the topic, the first paragraph is not about eating alone and should be deleted. Start with paragraph 2 (line 48) and only focus on older adults (your target population). Please explain in depth what is known from literature about the reasons people eat alone, what kind of research and what are the consequences. What are the mechanisms behind the presumed relation between eating alone and decreased/insufficient food intake and mental health. Do we know if eating alone when living together is worse compared to living alone completely? do we know if the people that choose to eat alone are better off compared to people that are forced to eat alone?

Response: Thank you very much for your constructive suggestion. We agree with the importance of focusing on the target population and showing what is already known in the literature in the Introduction. We reconstructed the Introduction section to incorporate previous study findings regarding older adults eating alone, the effect of eating alone, and the factors promoting healthy eating. However, we could not find a study exploring the reasons of eating alone nor one that focuses on the difference of willingness to eating alone. These points are partly covered in our study.

2. Methods: Please add more information and details on the sample population. Where did they come from, what were reasons to not participate? please add the semi-structured interview guideline. And more details on how you guided the interview. how many researchers were present during the interview? where is all info collected - add more extended table 1? Not all in table 1? Table 1: what is 'going out'? how were the questions asked? add answer categories if used.

Response: Thank you very much for your suggestion and questions. We agree that we have to show more detailed information in the Methods section. We have now elaborated figure 1 to show the sampling procedure with reasons for non-participation. To show how we conducted the interviews, we added more information concerning the interview format and added our interview guide as a supplemental file. Regarding table 1, we added new information such as the trigger(s) of eating alone and psychological status (scores of geriatric depression scale) and omitted unnecessary information. (p. 5, lines 82–85)

3. Update the part on data analyses with more details on the data handling and data saturation, how do you know that the number of interviews were enough to ensure that no new themes would emerge from additional data.

Response: Thank you very much for introducing the helpful article (Guest et al.: How many interviews are enough? An experiment with data saturation and variability. Field Methods, 2016). We reviewed it carefully and reconsidered the data saturation of our study. As shown in figure 1, our participants were selected from the participants in an existing population-based cohort study. Of the total sample, 51 met the criteria. Although we tried to recruit all of them, we could only conduct interviews with 20 of them. The nested study design limited the possibility to recruit more participants. We continuously analyzed the data and developed the themes during data collection. Although we ensured that a new theme was not extracted in the closing stage of data collection/analysis, there is a possibility that recruiting more participants would enrich the study results. We added this information to the data analysis (p. 6, lines 96–97) and Discussion sections (p. 16, lines 265–269).

4. Results: Please add more information on: who mentioned the specific themes, where answers given mainly by men or women, older younger older adults, where the quotes by older adults that choose to eat alone, or were forced to do so? Add analyses and cross-analyses, compare groups.

Response: Thank you for your suggestions. Based on the journal requirements, we removed the information of participants’ sex and age from table 1. Instead, we added more specific information regarding the psychological status and trigger(s) of eating alone for each participant. In the Results section, we added the numbers of participants who mentioned the theme and the common tendencies among them. Regarding participants’ feelings concerning eating alone, we were unable to conclude if participants were willing to eat alone from the data. This could be a result of “routinization,” which causes people to cease thinking about eating alone. (page 7, table 1)

5. For the themes: eating in solo-friendly, age related please provide other quotes and examples that really reveal the link between eating alone and the health consequence. Please extent figure 1 with more details.

Response: “Eating in a solo-friendly environment” and “age-related changes” have been recategorized as “background factors,” as we updated the conceptual model (please see figure 2). These two themes appear universally as people age; however, they seemed to be predictors of eating alone for some participants. As these themes are background factors of eating alone, it is difficult to show a clear link with health consequence by using quotes. We read through the transcript of the interviews again and selected other quotes which could more clearly show the link between these themes and eating alone (pages 8-9).

6. Discussion please elaborate to the part about stabilizers? it seems like this is a good thing, while you explain in the intro that eating alone has negative health consequences. Or is this different for people that are used to living alone/eating alone?

Response: We elaborated the section discussing the stabilizer as well as those discussing the triggers and background factors in the Discussion section. Eating alone is a simple behavior, but it has a risk of negative health consequences. Once those eating alone get used to this behavior, they could stabilize it as a daily routine. Thus, “routinization” plays the role of a stabilizer of eating alone. Regarding “eating in a solo-friendly environment,” we recategorized it as a background factor, rather than a stabilizer.

7. 217-221: was this a result from the interviews or is this a solution if there is no other choice?

Response: No, this is not from our data. We intended to show the environmental effect on individual eating alone behavior. We have reconstructed the Discussion section, please refer to the main manuscript.

8. What is the solution for the effect on mental health? eating together in community setting? please support with references and studies that show that this might help/improve mental status / quality of life?

Response: The effect of eating together is under investigation. A hospital-based study reported the effect of eating together on increasing energy intake among 48 older patients (Wright et al. J Hum Nutr Diet. 19(1): 23-6. 2006). Another cross-sectional study revealed the negative impact of low frequency of eating together on health outcomes among older adults living alone (Ishikawa et al. J Nutr Health Aging. 22(3): 341-353. 2018). Although more evidence is needed to conclude the positive effect of eating together on mental status/quality of life, studies are accumulating rapidly. We added this information to the Discussion section (p. 15, lines 252–255).

9. 234-237 support with more studies - RCTs available?

Response: Referring to our response to the previous comment, studies that show the positive effect of eating together is accumulating rapidly. However, RCTs are not available yet. We rewrote this part referring to existing literature (p. 15, lines 252–255).

10. 242-243 how can you conclude physiologically healthy participants? If you have this information, more details should be included in demographics and results section.

Response: As our participants were selected from an existing population-based cohort study, we have data regarding their psychological status at the time of conducting the cohort study. The participants’ psychological statuses were measured using the geriatric depression scale 15 and only three participants had scores of 10 or more, indicating “severe depression.” We added the participants’ scores on the geriatric depression scale to table 1 and mentioned them in the Results section (p. 6, lines 106–107) and the Discussion section (p. 16, lines 263).

11. 248-251: do people eat more when they take more time for their meals? please explain about study mentioned and support with other references.

Response: The article we cited reports that time spent on eating at home has been reduced in many countries (Warde et al., Acta Sociologica 2007;50(4): 363–385). This indicates that the present generation would think lightly of eating at home because they are busier with daily tasks than the past generation. We intended to show that there might be a change in the time taken for eating and its possible effect on our participants. To support this idea, we added a sentence and another citation (Cheng et al. Br J Sociol. 2008; 58(1): 39-61) (p. 16, line 273-276).

12. 263-265 this contradict the part on generalizability (lines 238/239) we don't know enough about the sample group to state something on generalizability or extension of results to other older adults in Japan.

Response: Thank you very much for your keen observation. We totally agree with the contradict expression and have deleted this sentence.

---

## [Decision Letter · Decision Letter 1]

27 May 2020

A qualitative study on the reasons for solitary eating habits of older adults living with family

PONE-D-19-29527R1

Dear Dr. Takahashi,

We are pleased to inform you that your manuscript has been judged scientifically suitable for publication and will be formally accepted for publication once it complies with all outstanding technical requirements.

With kind regards,

Heidi H Ewen, Ph.D.

Academic Editor

PLOS ONE

Additional Editor Comments (optional):

Reviewers' comments:

Reviewer's Responses to Questions

**Comments to the Author**

1. If the authors have adequately addressed your comments raised in a previous round of review and you feel that this manuscript is now acceptable for publication, you may indicate that here to bypass the “Comments to the Author” section, enter your conflict of interest statement in the “Confidential to Editor” section, and submit your "Accept" recommendation.

Reviewer #1: All comments have been addressed

Reviewer #3: All comments have been addressed

2. Is the manuscript technically sound, and do the data support the conclusions?

Reviewer #1: Yes

Reviewer #3: Yes

3. Has the statistical analysis been performed appropriately and rigorously? 

Reviewer #1: N/A

Reviewer #3: Yes

4. Have the authors made all data underlying the findings in their manuscript fully available?

Reviewer #1: Yes

Reviewer #3: Yes

5. Is the manuscript presented in an intelligible fashion and written in standard English?

Reviewer #1: Yes

Reviewer #3: Yes

6. Review Comments to the Author

Reviewer #1: The authors have fully answered to the questions and the revised article had improved to be understood well.

Reviewer #3: This study is very interesting and provides a good preliminary exploration of reasons for older adults in Japan to eat alone despite living with family members. I think that future studies could probe the participants' feelings about eating alone and link to additional health and nutrition data, such as percentage intake of estimated nutrient needs, variety of diet, and weight status or changes in weight.

7. PLOS authors have the option to publish the peer review history of their article (what does this mean?). If published, this will include your full peer review and any attached files.

Reviewer #1: No

Reviewer #3: Yes: Whitney Bignell, PhD, RDN, LD

---

## [Editor Report · Acceptance letter]

29 May 2020

PONE-D-19-29527R1 

A qualitative study on the reasons for solitary eating habits of older adults living with family 

Dear Dr. Takahashi:

I am pleased to inform you that your manuscript has been deemed suitable for publication in PLOS ONE. Congratulations! Your manuscript is now with our production department. 

With kind regards,

on behalf of

Dr. Heidi H Ewen 

Academic Editor

PLOS ONE